

# A review of the available cropland and land cover maps for South Asia

Prashant Patil

Remote Sensing & GIS Division, International Crops Research Institute for the Semi-Arid Tropics (ICRISAT), Patancheru, 502324, India

*Correspondence to*: Prashant Patil (ptpatil@hotmail.com)

**Abstract.** A lack of accuracy, uniqueness, and systematic classification of cropland categories, together with long-pending updates of cropland mapping, are the primary challenges that need to be addressed in developing high-resolution cropland maps for south Asia. In this review, we identify the major concerns, particularly the paucity of knowledge regarding the spatial distribution of major crop types within south Asia, which hinder policy and strategic investment and delay efforts to improve

food security for a rapidly growing human population at a time of constant market instability and changing global climate. We specifically focus on the various available land cover and cropland maps at national and regional scales in south Asia and at the global scale, and describe the classification methods and approaches used for satellite images of different spatial resolution at different scales. The overall focus of this paper is to review the need for timely updated high-resolution cropland maps of south Asia.

**Keywords:** Croplands, Land Cover, Spatial Resolution, Classification, South Asia

## 1 Introduction

South Asia is an essentially low-latitude tropical region, the inhabitants of which mainly rely on agriculture as a primary source of employment, food, and economy. In this region, the ongoing change in global climate has become a major concern for food security. In south Asia (Fig. 1), approximately 70% of the population lives in rural areas, wherein agriculture is the main source

of employment (ADB 2013). Agriculture generates approximately 18% of the south Asia's gross domestic product (GDP) and provides employment for 50% of the population (FAO 2008; FAO 2010; ADB 2013). Poverty in south Asia has constantly been at a high level, and the largest proportion of the world's poor, approximately 43%, resides in this region. Furthermore, according to the latest Global Hunger Index (GHI = 29.4 points; GHI 2015), the level of hunger in south Asia is the second highest in the world. Owing to the rapidly growing population, increasing amounts of agricultural land are being lost to the

spread of urban settlements in this region, and as a consequence, food security is gradually becoming a major burden and a key challenge for the scientific community, policy makers, and economists (Table 1). In the year 2050, the world population has been projected to exceed 9 billion, which represents a substantial increase from the over 7 billion in 2010 (Roberts 2011). Due to the continuously growing wealth of low- and middle-income developing countries, an increase in annual food production of 60%–70% will be required to feed their rapidly growing populations (FAO 2009; Alexandratos and Bruinsma

2012; ADB 2013).



Mapping cropland extent can yield timely, updated, and accurate cropland information that provides essential inputs to crop monitoring systems and early warning systems, such as CropWatch, Global Information and Early Warning System (GIEWS), the Early Warning Crop Monitor and the Famine Early Warning Systems Network (FEWSNET), and Forecasting Agricultural output using Space, Agrometeorological and Land-based observations (FASAL) (Wu et al. 2015; Hannerz and Lotsch 2008;

Vancutsem et al. 2012; Parihar and Oza 2006). Such mapping represents an important step in agricultural production assessment and has direct benefits for the early forecasting of crop type pattern distributions and spread of diseases, and also provides information for environmental climate change studies (Lobell, Bala, and Duffy 2006)

In the monitoring of croplands and climate change-related forecasting, cropland maps are used as a mask to separate cropland areas in order to assess and compare crop growth rates and conditions, and to study climatic change scenarios under different

agricultural land uses. Additionally, remote sensing technology provides an opportunity to monitor global cropland in a spatially clear, cost-effective, well-organized, and unbiased manner (Yu et al. 2013a; Gordon et al. 2005; Thenkabail et al. 2010; Foley et al. 2011; Gong 2011).

## 1.1 Croplands

Croplands are continuously monitored in near-real time by different organizations and research institutes at both national and

global scales. The Group on Earth Observations Global Agricultural Monitoring (GEOGLAM) works globally in collaboration with the agricultural departments of many countries, and the Food and Agricultural Organization of the United Nations (FAO) offers continuously updated cropland information on a global basis (FAOSTAT; http://faostat.fao.org/beta/en/#country/100).

## 1.2 Policy

At the national level, accurate and real-time cropland area and yield estimates are needed by the scientific community, policy

makers, and economists in order to facilitate appropriate planning and decisions regarding the quantity of food that should be stored, circulated, or distributed, and to enable an evaluation of food losses that occur due to various adverse factors along the food supply chain. Inappropriate evaluations of food availability may affect trade decisions due to unpredictable price fluctuations and food shortages. At national, regional, and global levels, detailed high-resolution cropland maps and cropland range datasets can help policy makers to identify where investment should be made by international bodies and national

governments to increase cropland productivity in the agricultural sector. At the regional and national levels, this information can be helpful for understanding the impacts of drought, application of chemical fertilizers and pesticides, and other natural and man-made disasters on cropland production (Funk and Brown 2009). At the national, sub-national, and farm levels, exact information on croplands can be used to study the factors affecting cropland productivity, and to evaluate the planning and appropriate management that may yield the projected results. Further, high-resolution accurate cropland mapping is also

required to study the multi-dimensional tasks associated with global ecological variation and the impacts of population growth and increased industrialization (Linda et al. 2015).

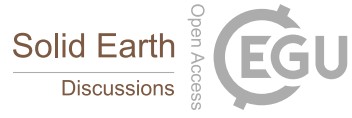

### 1.3 Yield gaps

To address the growing demand for food in the near future, there will need to be either an increase in cropland area or an increase in productivity, through new innovations, in areas that are currently experiencing yield gaps. There are also environmental impacts and transactions associated with both these paths that need to be studied to identify options for reducing adverse effects (Tilman et al. 2011). Nevertheless, it is generally believed that a large expansion of existing cropland would be detrimental in the long term, due to intensification, and therefore it is very important to establish when, where, and how to increase crop production through systematic planning and scientific investigations of existing cropland production areas (Tilman et al. 2002; Van Wart et al. 2013). Hence, having detailed and accurate information on the extent of existing available cropland, through having access to high-resolution cropland maps based on remote sensing technology, is important for conducting advanced research on croplands.

### 1.4 Data mismatches

Smith et al. (2010) examined the outputs of a different number of unified assessment models in relation to global climate change in crop-growing areas as well as other land-use types for the years 2020 and 2050 under abundant drivers of change. They found that many of the predictions for cropland change using these models are within the uncertainty level of the 300 Mha range for total cropland area estimates derived from global land cover mapping. Different studies have concluded that model outputs and analyses can vary substantially depending upon the land cover product and the spatial resolution of maps (Linard, Gilbert, and Tatem 2010; Quaife et al. 2008). However, the mismatches in the different available datasets on cropland extent and production are presently too high for use in various research applications. Recently, one study has reported that current global estimates of the total land under cropland differ by as much as 300 Mha. In this study, Fritz et al. (2011a) compared two global land cover maps, one of which showed a cropland coverage of approximately 1600 Mha, whereas the other showed a coverage of approximately 1300 Mha. This type of mismatch among different studies highlights the uncertainties in considering how other important factors of change in agricultural systems, such as biofuel production, increasing demand for livestock products, and the continuous expansion of urban areas due to rapid population growth and climate change, might affect food production. Case studies such as those mentioned above indicate that there is a need for improved high-resolution cropland maps at global, regional, national, and sub-national levels, which can be used at multiple scales for planning, monitoring, and assessment from the farm to sub-national, sub-national to national, national to regional, and regional to global scales (Linda et al. 2015).

This article reviews various studies that have been carried out at national, regional, and global scales for the mapping of land cover and croplands, and describes the classification methods and approaches used for satellite images of different spatial resolution at different scales. The overall focus of this paper is to review the need for timely updated high-resolution cropland maps for south Asia.



## 2 Currently available satellite-based global, regional, and national land-cover and cropland maps at different resolutions

The available literature suggests that changes in global land use and land cover are among the main reasons for global climate change (Foley et al. 2005; Ramankutty et al. 2008; Arino et al. 2008; Thenkabail et al. 2009; Bontemps et al. 2011).

Accordingly, there is currently a concerted effort to map land cover and its changes on a global scale. Land cover and cropland information is being made increasingly available, by virtue of the current rapid advances in remote sensing technology. This technology delivers an objective, frequent, and consistent measure of what the Earth's land cover looks like. There are, however, several limitations to remote sensing technology, e.g., in multispectral mode, the Earth cannot be observed through clouds, dust, and other atmospheric factors. Moreover, active global land cover maps, e.g., those based on Moderate Resolution

Imaging Spectroradiometer (MODIS) data, can at best only be obtained at relatively low spatial resolutions, e.g., 1 km$^2$, 500 m$^2$, and 250 m$^2$, which is inadequate for capturing the land-use patterns, cropland changes, and distribution of different crop types in some complex landscapes (Ozdogan and Woodcock, 2006). Until recently, global land cover products were developed using coarse-resolution satellite images; however, with the opening up of the Landsat archive in 2008, which is freely available at a spatial resolution of 30 m$^2$, globally higher resolution land cover maps are now beginning to appear (Wulder et al. 2012).

### 2.1 Currently available satellite-based global land cover and croplands maps

#### 2.1.1 International Geosphere Biosphere Programme Data and Information Systems (IGBP-DIS)

A global land cover map with a spatial resolution of 1 km$^2$ was derived via an unsupervised classification method using monthly 1-km$^2$ spatial resolution Advanced Very High-resolution Radiometer (AVHRR) data for the period 1992–1993. A total of 17 general land cover classes were defined, in which agricultural land has been classified in two different classes: croplands and cropland/other vegetation mosaic (Loveland et al. 2000).

#### 2.1.2 University of Maryland Global Land Cover (UMd-GLC)

A land cover classification map with a spatial resolution of 1 km$^2$ was derived using AVHRR satellite data. The land cover comprises 12 classes, within which agricultural area has been classified in a single class, i.e., croplands (Hansen et al. 2000).

#### 2.1.3 Global Land Cover SHARE (GLC-SHARE)

A land cover map with a spatial resolution of ~1 km$^2$ was derived at the global level. It provides a set of 11 major thematic land cover layers obtained from the combination of "best available" datasets, e.g., high-resolution national, regional, and/or sub-national land cover databases (Latham et al. 2014).



### 2.1.4 Global Land Cover map for the year 2000 (GLC2000)

A global land cover map with a spatial resolution of 1 km$^2$ was derived from daily data obtained by a SPOT-4 satellite. In this map, agricultural land has been classified in three different classes: cultivated and managed areas, mosaic-cropland/tree/cover/other natural vegetation, and mosaic-cropland/shrub and/or grass cover (Bartholome and Belward 2005).

### 2.1.5 MODIS-based MCD12Q1

A global land cover map with a spatial resolution of 1 km$^2$ was derived using the MODIS land cover classification algorithm (MLCCA), which employs a supervised methodology that adopts the C4.5 algorithm as the primary classification. Among a total of 17 major classes, agricultural land has been classified in two classes: croplands developed & mosaic lands and cropland/natural vegetation mosaics (Friedl et al. 2002).

### 2.1.6 Global Major Crops Distribution Map

A crop distribution map with a spatial resolution of ~10 km$^2$ was derived using a combination of satellite-derived land cover data and agricultural census data, providing a dataset of the global distribution of 18 major crops. The dataset is representative of the early 1990s, and describes the fraction of a grid cell that is occupied by each of the 18 crops worldwide. Although there has been research on how the major crop belts are formed worldwide by growing different crops in combination, this dataset is not particularly applicable to studies on a local scale. Nevertheless, the dataset can be used for the analysis of crop geography in a regional-to-global context. This dataset can also be used to determine the global patterns of farming systems in different agro-climatic zones for the study of food security, and to develop global ecosystem and climate models to predict the environmental consequences of cultivation of different crops (Leff, Ramankutty, and Foley 2004).

### 2.1.7 Agricultural Lands in the Year 2000 (M3-Cropland and M3-Pasture Data)

Land-use datasets with a spatial resolution of ~10 km$^2$ were derived by combining national-, state-, and county-level census statistics using a global dataset of croplands. The resulting land-use datasets describe the area (harvested) and yield of 175 distinct crops for the year 2000. By aggregating individual crop maps, fresh maps were produced for 11 major crop groups, crop net primary production, and the following four physiologically based crop types: annuals/perennials, herbaceous/shrubs/trees, C3/C4, and leguminous/non-leguminous crops (Monfreda, Ramankutty, and Foley 2008).

### 2.1.8 Collection 5 MODIS Global Land Cover Type product

A global land cover map with a spatial resolution of 500 m$^2$ was derived with an increased spatial resolution compared to the Collection 4 MODIS Global Land Cover Type product (the previous version) (Friedl et al. 2002). Additionally, many parameters of the classification algorithm have been modified by reviewing the training site database and by adding the land surface temperature as an additional input feature. Furthermore, the use of ancillary datasets in the post-processing of joint

decision tree outcomes has also been updated. Here, agriculture land has been classified in two classes: croplands and cropland/natural vegetation mosaics (Friedl et al. 2010).

### 2.1.9 Global Land Cover by National Mapping Organizations (GLCNMO)

A global map with a spatial resolution of 1 km$^2$ was derived by using eight periods of 16-day composite 7-band 1 km$^2$ MODIS data for the year 2003. The dataset contains approximately 20 land cover classes, which were defined using the Land Cover Classification System. Among these classes, 14 classes were derived using supervised classification. Agricultural land is classified in three classes: cropland, paddy field, and cropland/other vegetation mosaic (Tateishi et al. 2011).

### 2.1.10 The Centre for Sustainability and the Global Environment (SAGE)

Global datasets of the distributions of croplands and pastures circa 2000 with a spatial resolution of ~10 km$^2$ were derived by combining agricultural inventory data and satellite-derived land cover classification datasets. These land cover classification datasets have been generated using highly detailed agricultural inventory data, which were obtained by using a combination of two different satellite-derived products, i.e., Boston University's MODIS-derived land cover product (BU-MODIS) and the GLC2000 dataset, and also by including the statistical confidence intervals on estimates. Agricultural land has been classified into two classes: cropland and pasture (Ramankutty et al. 2008).

### 2.1.11 History Database of the Global Environment (HYDE)

A global cropland database with a spatial resolution of ~10 km$^2$ was derived using historical datasets collected over the past 12,000 years, with a consideration of the basic demographic and agricultural driving factors. By combining the historic population data and cropland and pasture statistics with satellite information and specific allocation algorithms that can change over time, spatially explicit maps have been derived, which are completely reliable and cover the period 10,000 BC to AD 2000. Agricultural land has been classified into two classes: cropland and pasture (Goldewijk et al. 2010).

### 2.1.12 Global Irrigated Area Map (GIAM)

A global irrigated area map with a spatial resolution of 10 km$^2$ was derived using multiple satellite datasets, Google Earth, and ground truth data. A single mega-file data-cube (MFDC) of the world with a total of 159 layers, corresponding to the hyperspectral dataset, was generated by the process of re-sampling different data types into a common 1-km$^2$ spatial resolution. The MFDC was then segmented based on elevation, temperature, and precipitation zones. A global total of 28 irrigated cropland classes were accordingly obtained (Thenkabail et al. 2009).

### 2.1.13 Global Map of Rainfed Cropland Areas (GMRCA)

A global rain-fed cropland area map with a spatial resolution of 1–10 km$^2$ was derived using remote sensing data. The final outcome is an aggregated nine-class GMRC, and country-by-country rain-fed area statistics have also been calculated (Biradar et al. 2009).

### 2.1.14 IIASA-IFPRI cropland percentage map

A global cropland percentage map with a spatial resolution of 1 km$^2$ was derived by combining individual cropland maps from global to regional to national scales, using the year 2005 as a baseline. The complete map products, together with the existing global land cover maps, such as GlobCover 2005 and MODIS v.5, regional maps, such as AFRICOVER, and national maps, can be procured from mapping agencies and other mapping organizations. All products are classified at the national level using

crowd-sourced data from Geo-Wiki to derive a map that resembles and reflects the likelihood of cropland. Validation of the newly derived IIASA-IFPRI cropland maps has been performed using very high-resolution satellite imagery via Geo-Wiki with an overall accuracy of 82.4% (Fritz et al. 2015).

### 2.1.15 Monthly Irrigated and Rain-Fed Crop Areas around the year 2000 (MIRCA2000)

Global maps with a spatial resolution of ~9.2 km$^2$, at a global-scale that is compatible with agricultural land-use practices,

were derived for monthly irrigated and rain-fed cropland areas around the year 2000. The basic approach followed here was initially to define cropping periods and growing areas for 402 spatial units and then to downscale the information to a grid cell. MIRCA2000 provides agriculture classes under both irrigated and rain-fed crop areas of 26 crop classes for each month of the year (Portmann, Siebert, and Doll 2010).

### 2.1.16 GlobCover Global Land Cover Map

A global land cover map with a spatial resolution of 300 m$^2$ was derived by using MEdium Resolution Imaging Spectrometer (MERIS) satellite data. The map has been classified by using different approaches in which the sub-system produces a global land cover map from cloud-free mosaics, and the classification runs separately for 22 equal-reasoning areas. The map has been classified into 23 different classes, among which agriculture has been classified into four classes: post-flooding or irrigated croplands, rain-fed croplands, mosaic cropland/vegetation (grassland, shrub land, and forest), and mosaic vegetation

(grassland, shrub land, and forest)/cropland (Arino et al. 2008; Bontemps et al. 2011).

### 2.1.17 Global Cropland Extent Map

A global cropland extent map with a spatial resolution of 250 m$^2$ was derived by using MODIS data. A set of 39 multi-year MODIS metrics, including four MODIS bands, Normalized Difference Vegetation Index (NDVI), and thermal data, has been



used to determine cropland phenology over the study period. Agricultural land has been classified in a single class, i.e., cropland (Pittman et al. 2010).

### 2.1.18 A Unified Global Cropland Layer

A combined global cropland map with a spatial resolution of 250 m$^2$ was derived for the year 2014. The map has been produced by combining the fittest products, followed by very inclusive identification and collection of national to global land cover maps. In this process, a multi-criteria analysis was performed at the country level to identify the priority areas for mapping of croplands (Waldner et al. 2015).

### 2.1.19 Finer Resolution Observation and Monitoring of Global Land Cover (FROM-GLC)

Global land cover maps with a 30-m$^2$ spatial resolution were derived using Landsat Thematic Mapper (TM) and Enhanced Thematic Mapper Plus (ETM+) datasets. The datasets were generated using a total number of 91,433 training points and 38,664 test samples collected via human interpretation of TM/ETM+ images. Four datasets of global land cover maps were derived by using four types of supervised classifiers, including the conventional maximum likelihood classifier (MLC), the J4.8 decision tree classifier, the random forests ensemble classifier (RF), and the support vector machine. Agricultural land has been classified in a single class, i.e., agriculture (Gong et al. 2013).

### 2.1.20 Fine Resolution Observation and Monitoring of Global Land Cover (FROM-GLC-seg)

A global land cover map with a 30-m$^2$ spatial resolution was derived using Landsat TM and ETM+ data. By following a segmentation-based approach to the Landsat imagery, MODIS data of spatial resolution 250 m2 has been downscaled along with the auxiliary data for 1 km2 to the segment scale based on TM data. Agricultural land has been classified into four classes: cropland (paddy rice, greenhouse, and others), orchard, managed grasslands, and temporally bare croplands (Yu, Wang, and Gong 2013b).

### 2.1.21 Global Land Cover (GLC)

A global land cover map with a spatial resolution of 30 m$^2$ was derived by using Landsat TM and ETM+ data following an approach based on the combination of pixel- and object-based methods with knowledge (POK-based), and by using supervised and decision tree classification methods. Agricultural land has been classified in a single class, i.e., cultivated land (Chen et al. 2015).



## 2.2 Currently available regional land cover and cropland maps in Asia

### 2.2.1 Southeast Asia

A regional land cover map of peninsular southeast Asia with a spatial resolution of 500 $m^2$ was derived by using daily MODIS satellite data (acquired January 2–July 3, 2010). Classification was performed using an unsupervised classification method, in which a combination of peatland maps, elevation information, and Daichi-Advanced Land observation satellite mosaic datasets were used to derive the final map. The land cover map covers the Malaysian Peninsula; the major islands of Sumatra, Java, Borneo, Sulawesi, and Mindanao; the western part of New Guinea; and numerous smaller islands of southeast Asia. Agricultural land has not been classified into any separate class but has been included within four other subclasses: Lowland mosaic, Montane mosaic, Lowland open, and Montane open (Miettinen et al. 2016).

### 2.2.2 South Asia

A rice/paddy map of south Asia with a spatial resolution of 500 $m^2$ was derived by using MODIS satellite imagery and different classification approaches, i.e., spectral matching techniques, decision trees, and ideal temporal profile databanks (Gumma et al. 2011).

### 2.2.3 South, Southeast, and East Asia

Rice maps of south, southeast, and east Asia have been derived by using information based mainly on statistical bulletins of the various countries, mostly dating from 1993 to 1996. The data are presented in tabular form and provide considerable detail at the district, division, and township levels. The classification used is as follows: dryland rice, deep-water rice, irrigated wet season rice, irrigated dry season rice, shallow rain-fed rice areas with water depth in the 0–30-cm range, and intermediate rain-fed rice, subject to greater water depths of 30–100 cm. For most countries, the area of each division is shown, as well as the total rice production and total population (Huke and Huke 1997).

### 2.2.4 South and Southeast Asia

Paddy rice maps with a spatial resolution of 500 $m^2$ were derived by using the 8-day composite images of MODIS satellite images obtained by the NASA EOS Terra satellite for the year 2002. Paddy rice fields are categorized by an early period of flooding and transplanting, during which there is a co-occurrence of surface water and rice seedlings. The maps were derived by applying a paddy rice mapping algorithm that uses the time series of MODIS-derived vegetation indices to identify the initial period of flooding and transplanting in paddy rice fields, based on the increased surface moisture (Xiao et al. 2006).

### 2.2.5 Northeast Asia

An annual paddy rice map for northeast Asia (in which rice is predominantly grown using a single cropping system) with a spatial resolution of 30 $m^2$ was derived using the Landsat 8 images. To process all the available Landsat 8 imagery of the year





2014, a cloud computing approach was adopted using the Google Earth Engine (GEE) platform. The study shows that the Landsat 8, GEE, and improved pixel-based paddy rice mapping algorithm can effectively support the yearly mapping of paddy rice in northeast Asia with high accuracy (Dong et al. 2016).

### 2.2.6 Central Asia

A regional land cover map of central Asia with a spatial resolution of 250 m$^2$ was derived by using MODIS time-series images for the years 2001 and 2009. The classification has been implemented by using the C5.0 algorithm based on seasonal features, which facilitates analyses of possible land cover and land-use changes. Agricultural land has been classified within two classes: rain-fed and irrigated agriculture (Klein, Gessner, and Kuenzer 2012).

### 2.3 Currently available satellite-based national land cover and cropland maps within south Asia

### 2.3.1 Afghanistan

A land cover map with a spatial resolution of 20 m$^2$ was derived using FAO/GLCN methodology and tools. The main data sources include satellite imagery from SPOT-4 (2009–2011) and Global Land Survey (GLS-2011) Landsat satellites, high-resolution satellite imagery, very high-resolution aerial photographs, and ancillary data. The national level legend has been organized by using the Land Cover Classification System (LCCS). FAO's Mapping Device Change Analysis Tools

(MADCAT) software has been used to create the database using object-based classification methodology. The full resolution land cover legend has 25 classes. To refine the interpretation, high-resolution images from various sources are used. A total of 25 original land cover classes have been aggregated into 11 generalized and self-explanatory classes. Agricultural land has been classified in two classes (ALCM, 2012) (http://www.glcn.org/activities/afg_lc_en.jsp).

### 2.3.2 Bhutan

The International Centre for Integrated Mountain Development (ICIMOD) has completed mapping of the land use/land cover of Bhutan at a spatial resolution of 180 m$^2$ by acquiring WiFs satellite data for the period 1996–1999. For generating the training samples, a land use/land cover map of Bhutan for the year 1994 has been used. This map was produced using Landsat-TM satellite data at 30 m$^2$ spatial resolution, which has been systematically field verified. Training samples within different major land use/land cover classes for digital classification of images have been generated by resampling of the available 1994

land use/land cover map using WiFs data at a spatial resolution of 180 m$^2$, and different land cover has been identified within the area of interest. Training sample points with different land cover codes were converted to imagine format and used for the classification, in which a maximum likelihood classifier has been applied for image classification. Of the 13 major land cover classes defined using this approach, agricultural land has been classified as a single class, i.e., agriculture (Sushil Pradhan, 2002).



### 2.3.3 Bangladesh

Seasonal paddy rice maps with a spatial resolution of 500 m$^2$ for the three seasons boro (December/January–April), aus (April/May–June/July), and aman (July/August–November/December) in Bangladesh were derived for the year 2000 by using MODIS NDVI maximum value composite data. Image classification was performed using an unsupervised classification

method, whereas the grouping of similar classes was performed using decision tree classification algorithms and a spectral matching technique. In mapping the extent of the rice crop in Bangladesh, seasonal district-level rice area statistics were used to assess the accuracy of the rice area estimates (Gumma et al. 2014).

### 2.3.4 India

A national-level agriculture land cover-type map of India with a spatial resolution of 56 m$^2$ was derived for the periods 2005–

2006 and 2011–2012 for the study of spatial and temporal variability in agricultural land cover types during the aforementioned periods, using multidate and multispectral Advanced Wide Field Sensor (AWiFS) data from Resourcesat-1 and 2. The map was generated using an 18-fold classification system, and rule-based techniques along with a maximum-likelihood algorithm were used for classification of land cover information as well as changes occurring within the agricultural land cover classes. Six agricultural land classes were identified: Kharif (June–October), Rabi (November–April), Zaid (April–June), area sown

more than once, fallow lands, and plantation crops (Sreenivas et al. 2015).

### 2.3.5 Nepal

ICIMOD has completed mapping of the land use/land cover of Nepal at a spatial resolution of 180 m$^2$. These maps were based WiFs satellite data acquired for the period 1996–1999 by generating sample points within different major land use/land cover classes for digital classification of images. Image classification was based on a supervised classification method and

application of a maximum likelihood classifier. Within the 13 major land cover classes, agricultural land has been classified in a single class, i.e., agriculture (Sushil Pradhan, 2002).

### 2.3.6 Pakistan

MODIS 250-m$^2$ spatial resolution data obtained on a daily basis and SPOT VGT 1000-m$^2$ spatial resolution data obtained on a 10-daily basis have been used to monitor the growth of crops. Multi-date SPOT-5 high-resolution datasets are also being

acquired during each cropping season for assessment of land surface changes and size of cropped areas through image classification into different crop types and land cover classes, which has facilitated crop monitoring and forecasting (PAK-SCMS, 2015).



## 3 Discussion

A lack of details has led to the highly divergent estimates that one finds in the aforementioned land cover and cropland maps. Worldwide, cropland distribution estimates derived from GlobCover are more than 20% higher than those derived from MODIS (Fritz et al. 2011a, 2011b; Linda et al. 2015). These differences can be attributed to a number of factors, including the use of different classification algorithms with considerably diverse parameters, diverse satellite datasets used for different algorithms, dissimilar spatial resolutions, and the different temporal windows used to develop the land cover and cropland maps (Linda et al. 2015). The land cover and cropland maps used for geospatial modeling can therefore have a theoretically huge influence on the outputs. Because of the vast areas of diverse types of cropland and due to the small land-holding capacity in the agricultural lands of south Asia, more attention should be focused on high- and very high-resolution cropland mapping that is updated on an annual or more regular basis. Recently, high- and very high-resolution satellite images, e.g., those obtained by Landsat 8 and Sentinel 1, 2, and 3, have become freely available. Accordingly, any long-term solutions should look to benefit from these data-rich information streams and seek to exploit modern technology and the availability of user-friendly tools for processing huge numbers of satellite images within a short period of time.

## 4 Conclusion

Before commencing cropland mapping, it essential to establish a suitable definition of cropland that is compatible with other cropland definitions, e.g., that of the FAO. This is because, in different countries, the cropland or agricultural classes that are often used to describe different classes in land cover maps are not clearly defined, and there is no common agreement as to what they constitute, Hence, establishing a suitable definition of cropland will assist in the mapping of all the classes that come under the category of cropland/agricultural class. Moreover, in future, this may also facilitate the sharing of data, thereby helping to ensure no large gaps in crop coverage or mismatches with other maps that are available globally.

With the availability of high-quality accurate maps produced on a regularly updated basis at the spatial and temporal resolutions of land cover and croplands, policy makers at global, regional, national, and sub-national levels will be able to make considerably better decisions for reliably assessing land-use policies and exploring long-term sustainability replacements. Assessing the impacts of climate change on croplands, food production, and environments, as well as the provision of other ecosystem services, and making appropriate decisions for the overall improvement of the agricultural sector, supported by joint public and private investment, will make a substantial contribution to improving food security in the south Asian region.

## Acknowledgments

Author is thankful to anonymous reviewers for their valuable comments and suggestions.



**Disclosure statement**

No potential conflict of interest.

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



**Table 1.** Trends in different classes of agriculture land area in south Asian countries from 1962 to 2012.

| Countries | Classes | 1962 (1000 ha) | 1972 (1000 ha) | 1982 (1000 ha) | 1992 (1000 ha) | 2002 (1000 ha) | 2012 (1000 ha) |
|---|---|---|---|---|---|---|---|
| India | Arable land | 156,700 | 160,186 | 163,246 | 162,706 | 160,432 | 156,542 |
| | Permanent crops | 5,700 | 4,800 | 5,500 | 7,300 | 9,600 | 12,800 |
| | Permanent meadows & pasture | 14,082 | 12,960 | 12,025 | 11,299 | 10,528 | 10,296 |
| | Agriculture land | 176,482 | 177,946 | 180,771 | 181,305 | 180,560 | 179,642 |
| Afghanistan | Arable land | 7,700 | 7,910 | 7,910 | 7,910 | 7,678 | 7,785 |
| | Permanent crops | 60 | 136 | 144 | 120 | 75 | 125 |
| | Permanent meadows & pasture | 30,000 | 30,000 | 30,000 | 30,000 | 30,000 | 30,000 |
| | Agriculture land | 37,760 | 38,046 | 38,054 | 38,030 | 37,753 | 37,910 |
| Sri Lanka | Arable land | 577 | 822 | 857 | 905 | 936 | 1300 |
| | Permanent crops | 945 | 1,084 | 1,000 | 1,000 | 980 | 1,000 |
| | Permanent meadows & pasture | 185 | 439 | 439 | 439 | 440 | 440 |
| | Agriculture land | 1,707 | 2,345 | 2,296 | 2,344 | 2,356 | 2,690 |
| Bangladesh | Arable land | 8,597 | 9,133 | 9,104 | 8,609 | 8,253 | 7,678 |
| | Permanent crops | 280 | 258 | 274 | 340 | 500 | 830 |
| | Permanent meadows & pasture | 600 | 600 | 600 | 600 | 600 | 600 |
| | Agriculture land | 9,477 | 9,991 | 9,978 | 9,549 | 9,353 | 9,120 |
| Pakistan | Arable land | 30,690 | 30,340 | 33,140 | 29,920 | 31,220 | 30,240 |
| | Permanent crops | 150 | 165 | 369 | 460 | 664 | 823 |
| | Permanent meadows & pasture | 5,000 | 5,000 | 5,000 | 5,000 | 5,000 | 5,000 |
| | Agriculture land | 35,840 | 35,505 | 38,509 | 35,380 | 35,884 | 36,063 |
| Nepal | Arable land | 1,806 | 2,055 | 2,291.60 | 2,327.30 | 2,335 | 2,118 |
| | Permanent crops | 25 | 27 | 29 | 37.9 | 120 | 208 |
| | Permanent meadows & pasture | 1,722 | 1,740 | 1,786 | 1,793 | 1,786 | 1,795 |
| | Agriculture land | 3,553 | 3,822 | 4,106.60 | 4,158.20 | 4,241 | 4,121 |
| Bhutan | Arable land | 100 | 110 | 135 | 135 | 111 | 100.2 |
| | Permanent crops | 13 | 16 | 18 | 19 | 19 | 12.4 |
| | Permanent meadows & pasture | 250 | 259 | 265 | 350 | 405 | 407 |
| | Agriculture land | 363 | 385 | 418 | 504 | 535 | 519.6 |
| Maldives | Arable land | 2 | 3 | 3 | 3 | 3 | 3.9 |
| | Permanent crops | 2 | 2 | 4 | 4 | 8 | 3 |
| | Permanent meadows & pasture | 1 | 1 | 1 | 1 | 1 | 1 |
| | Agriculture land | 5 | 6 | 8 | 8 | 12 | 7.9 |

(Source: FAOSTAT, 2012)





**Fig. 1**. Map showing the south Asian countries (Afghanistan, Bangladesh, Bhutan, India, Sri Lanka, Maldives, Nepal and Pakistan).

