# Peer review of "A review of the available cropland and land cover maps for South Asia"

_Solid Earth, 2017_

## Referee Comment (RC1) · Anonymous Referee #1 · 9 Nov 2017

This manuscript listed lots of LUCC datasets and cropland maps of South Asia. But I couldn't find any helpful results or discussions. The manuscript did not propose any new techniques or present any significant results that could be useful to other land cover studies. A simple list of LUCC datasets and cropland maps is not enough for a paper at the level of the journal.

There are some suggestions for further revision:

1. In those LUCC datasets and cropland maps, which one is the most suitable for South Asian, and why?

2. Fig.1 is meaningless just displayed the boundaries of South Asian. I think a LUCC map of South Asian should be better.

[Figure]

3. There is repetition on the content, especially in section 2.2 and 2.3. National land cover and cropland maps in 2.3 are just subsets of globe of regional datasets and maps. This section is unnecessary.

4. The unit of spatial resolution of remote sensing data should be the length unit, not the area unit.

5. The name of LUCC dataset in 2.1.21 should be GlobeLand30.

---

## Author Comment (AC1) · 9 Nov 2017

Referee comment: This manuscript listed lots of LUCC datasets and cropland maps of South Asia. But I couldn't fiĄnd any helpful results or discussions. The manuscript did not propose any new techniques or present any signifiĄcant results that could be useful to other land cover studies. A simple list of LUCC datasets and cropland maps is not enough for a paper at the level of the journal.

Authors reply: The main objective of the article is to list the available land cover and cropland maps of the South Asian region and to highlight the methodology adopted in mapping the land cover and cropland maps. To know and understand, a lack of accuracy, uniqueness, and systematic classification of cropland categories, together with

long-pending updates of cropland mapping, are the primary challenges and that needs to be addressed in developing high-resolution cropland maps for the South Asian region by ensuring and adopting the unique definition of cropland categories which can be applied globally. Which can be helpful; • In continuous monitoring of croplands with higher accuracy, and for policymakers to know and understand the at the regional and national levels. • Further, high-resolution accurate cropland mapping is also required for policymakers to study the multi-dimensional tasks associated with global ecological variation and the impacts of population growth and increased industrialization on croplands. • To know and to understand to overcome the yield gaps and importantly the uncertainties associated with data mismatch.

Suggestions by a referee for further revision:

Referee suggestion: 1. In those LUCC datasets and cropland maps, which one is the most suitable for South Asian, and why?

Authors reply: The main objective of the listing the available land cover & cropland datasets are to know presently, how many datasets are available for the South Asian region and within a South Asian region, there resolutions, the year of mapping and the methodology adopted in mapping and how many cropland categories available. All available land cover & cropland maps are important for South Asian region to know and understand the major concerns, particularly the paucity of knowledge regarding the spatial distribution of major crop types within South Asia, and the knowledge in further proposing & developing the methodology for mapping the high-resolution cropland maps for the South Asian region by ensuring and adopting the unique definition of cropland categories which can be applicable globally.

Referee suggestion: 2. Fig.1 is meaningless just displayed the boundaries of South Asian. I think an LUCC map of South Asian should be better.

Authors reply: Thank you for your valuable suggestion. I will replace with available LUCC map of South Asia.

Referee suggestion: 3. There is repetition on the content, especially in section 2.2 and 2.3. National land cover and cropland maps in 2.3 are just subsets of globe of regional datasets and maps. This section is unnecessary.

Authors reply: Section 2.2 and 2.3 are different and not the subsets of a globe of regional datasets and maps. Section 2.2 and 2.3 are important sections which highlight the specific regional level and country level individual studies and maps. Section 2.2 Deals within continent level studies carried out and mainly highlights the specific region wise available land cover & croplands maps within Asia. i.e. • A regional land cover map of peninsular Southeast Asia (Miettinen et al. 2016). • A rice/paddy map of South Asia (Gumma et al. 2011). • Rice maps of South, Southeast, and East Asia (Huke and Huke 1997). • Paddy rice maps of South and Southeast Asia (Xiao et al. 2006). • An annual paddy rice map for northeast Asia ((Dong et al. 2016). • A regional land cover map of central Asia ((Klein, Gessner, and Kuenzer 2012). Section 2.3 Deals within sub-continent level studies and mainly highlights the specific country wise land cover and cropland maps within South Asia. i.e. • The land cover map of Afghanistan. (ALCM, 2012) (http://www.glcn.org/activities/afg_lc_en.jsp). • The land use/land cover map of Bhutan (Sushil Pradhan, 2002). • The seasonal paddy rice maps of Bangladesh (Gumma et al. 2014). • A national-level agriculture land cover-type map of India (Sreenivas et al. 2015). • The land use/land cover map of Nepal (Sushil Pradhan, 2002). • The land use/land cover map of Pakistan (PAK-SCMS, 2015).

Referee suggestion: 4. The unit of spatial resolution of remote sensing data should be the length unit, not the area unit. Authors reply:

Thank you for your valuable suggestion. I will replace all spatial resolution of remote sensing data in manuscript with length unit.

Referee suggestion: 5. The name of LUCC dataset in 2.1.21 should be GlobeLand30.

Authors reply: Thank you for your valuable suggestion. I will change the name of

LUCC dataset in section 2.1.21 with GlobeLand30.

Please also note the supplement to this comment:
https://www.solid-earth-discuss.net/se-2017-121/se-2017-121-AC1-supplement.pdf

---

## Referee Comment (RC2) · Anonymous Referee #2 · 14 Dec 2017

General comments

The manuscript is listed as a review paper but it only provides a list of the current available cropland and land cover maps. Although I understand the need and value of such manuscript, the manuscript lacks a clear scientific question (an overall focus is not a scientific question). As a result, the manuscript has a short and weak discussion and conclusion sections. Nevertheless, I believe that the topic of the manuscript is relevant and is suitable for the Solid Earth journal.

Suggestions for further major revision on the manuscript

- Present a stronger research question and objectives for the manuscript. It can be followed by a methodological approach (such as review the LUCC for South Asia) to

answer the research question.

- I believe that a more concise section of the available datasets can be achieved. For example, presenting the available datasets in a table. This can make it easy to compare different datasets and identify weaknesses in the discussion.

- Insert useful figures and tables. Fig.1 doesn't add any relevant information in a scientific manuscript. Also about Fig.1: if there is the need to keep such figure, work on a better labelling of the figure to avoid the issue with the labels of Maldives and Sri Lanka.

- Identify the authors of the current available datasets. The text is not clear if it was the author of the manuscript executing all datasets (the way the text is written in English).

- In the section "Introduction" there are some references missing to support strong statements.

Other minor technicalities

- Pag.1, line 26. Replace "Owing" by "Due".

- Pag.1, line 26 to 28. Even with the table it is a strong argument that should be supported by more references.

- Pag.2, line 12. Consider remove the sub-section "Croplands" or develop a more 2 sentences.

- Pag.2, line 19 to 23. Consider make shorter sentences to avoid a fragmented text for the reader. Also missing some references.

- Pag.3, line 2 to 3. Reference missing.

- Pag.3, line 5. Consider change the expression "generally believed" or justify it with references.

- Pag.3, line 19. Replace "estimates" by "estimations".

- Pag.4, line 7. Consider change the expression "looks like".

- Pag.4, line 17. Consider add reference to the "unsupervised classification method".

- Pag.6, line 1. Replace "agriculture" by "agricultural".

- Pag.9, line 3. Example of possible confusion with the author of the datasets in the use of the verb "was derived".

- Pag.12, line 15. Consider replace "it essential" by "it is essential".

- Pag.12, line 18. Consider replace the first comma for a final stop (before "Hence").